# Indicators of Sustainable Forestry: Methodological Approaches for Impact Assessments across Swedish Forestry

Eskil Mattsson [1,2,*](ORCID), Per Erik Karlsson [1], Martin Erlandsson [1,3], Åsa Nilsson [1] and Hampus Holmström [4]

[1] IVL Swedish Environmental Research Institute, Aschebergsgatan 44, 411 33 Gothenburg, Sweden; pererik.karlsson@ivl.se (P.E.K.); martin.erlandsson@ivl.se (M.E.); asa.nilsson@ivl.se (Å.N.)
[2] Gothenburg Global Biodiversity Centre (GGBC), 405 30 Gothenburg, Sweden
[3] Department of Civil and Architectural Engineering, KTH Royal Institute of Technology, 100 44 Stockholm, Sweden
[4] Department of Forest Resource Management, Swedish University of Agricultural Sciences, 901 83 Umeå, Sweden; hampus.holmstrom@slu.se
[*] Correspondence: eskil.mattsson@ivl.se

**Abstract:** Approaches for evaluating integrated sustainability impacts in forest management enable the harmonization of environmental, social, and economic considerations. Here, we present a methodological framework for quantifying and balancing impacts on widely different aspects of sustainability of different future scenarios for forestry in managed forests in Sweden. The method includes indicators for impacts on climate change, biodiversity, and social and economic values. The indicators were normalized to a standardized scale using reference scenarios and target values. The proposed method was applied for three different future scenarios for forestry over a 100-year period in two different counties in southern and northern Sweden, respectively. The results show the importance of evaluating indicator performance in forestry across diverse regions of the country and tailoring assessments of individual forest owners to their specific local conditions. Long-term assessments are also crucial due to the varying impacts of indicators over time. The methodology requires continuous refinement and can be used as a basis for disclosing the environmental performance of a product based on forest raw materials. It also facilitates the assessment of sustainability in alternative future forestry scenarios and is adaptable to other countries with comparable forestry and forest characteristics.

**Keywords:** forestry; sustainability; indicators; biodiversity; climate change; life-cycle assessment; wood production; bioeconomy

## 1. Introduction

Forest ecosystems play a pivotal role in mitigating climate change and reducing biodiversity loss [1,2]. They are significant not only as carbon sinks but also as environments for recreation and physical activity, thereby promoting overall well-being. This backdrop underscores the urgent need for sustainable forestry practices globally [3,4]. In the forestry sector, the concept of sustainability has evolved from a narrow focus on sustainable wood production to a broader assessment to encompass environmental, social, and economic sustainability across entire value chains. Sustainability impact assessments, in this respect, might represent a proactive and holistic evaluation method, aimed at equilibrating the potential consequences of policy actions and facilitating life-cycle assessments of forest industry products [5]. However, these assessments are extremely complex and encompass a variety of different aspects. Hence, there is an urgent need for methods and tools that enable integrated assessments, balancing environmental, social and economic impacts and analysing synergies and trade-offs between management goals [6].

There have been previous efforts to develop tools for sustainability impact assessments of forestry and forest product value chains at the global, national, and regional

levels [5,7]. The Tool for Sustainability Impacts Assessment, ToSIA, was developed for the sustainability impact assessment of forest wood chains [8–10]. One strength of TOSIA was its ability to offer a framework for analyzing the impacts of new policies, shifts in market conditions, management decisions, or emerging technologies for the sustainability of forestry value chains. However, its focus was predominantly on the impacts of the value chains downstream from forestry. The Landscape simulation and Ecological Assessment (LEcA) tool was developed to analyze synergies and trade-offs among ecosystem services in different forest management scenarios [11]. Tested in Kronoberg county, southern Sweden, it examined two forestry scenarios in the whole region: the first predominantly featured even-aged forestry (EAF), and the second combined continuous-cover forestry (CCF) with intensified EAF. The tool effectively illustrated the trade-offs between industrial wood and bioenergy production versus the maintenance of habitat, recreation, and carbon storage. Additionally, the Forest Europe process developed pan-European criteria and indicators for sustainable forest management consisting of 6 criteria and 35 quantitative indicators, though trade-offs between, for example, environmental and societal interests were not explicitly addressed [12,13].

In Sweden, forestry plays a vital role in the national economy [14,15] and is aligned with national environmental quality objectives [16]. Every year, all environmental objectives, including those related to Sustainable Forests, are followed up. Additionally, an in-depth evaluation of the conditions for achieving these objectives is conducted at least once every four years. Over the past 90 years, the forest stock has more than doubled [17,18]. The forest resource predominantly consists of native species, with Scots pine (*Pinus sylvestris*), Norway spruce (*Picea abies*), and European silver birch (*Betula pendula*) being the dominant tree species. Despite constituting less than one percent of the world's forested land, Sweden ranks as the fourth-largest global exporter of pulp, paper, and sawn wood products [19]. Production volumes in 2022 reached 11.8 million tonnes of pulp, 8.5 million tonnes of paper, and 19 million cubic meters of sawn timber [20]. Moreover, discussions regarding Swedish forestry are characterized by strong polarization and conflicts. These discussions balance economic returns in the form of produced forest raw materials and their utilization in various value chains with the conservation of biodiversity, climate benefits from increased carbon storage in forests and forest-industrial products, and the recreational value of forests [21–23]. It is evident that the impact of forestry on all aspects of sustainability will not be fully optimized [24,25]. This underscores the need for the standardization of methods and target values to assess the sustainability of Swedish forestry, in broad dialogue with various stakeholders both nationally and internationally. The increasing consumer demand for environmental performance disclosures also drives the need to quantify, compare, and link various aspects of sustainability to the production of forest resources and their products [26].

In this study, we present a methodological framework for performing integrated sustainability impact assessments using a diverse range of indicators for managed forests in Sweden. The methodology is flexible and intended to be continuously revised as new insights are gained. The hypothesis was that the results from the integrated sustainability impact assessments of Swedish forestry will vary considerably both temporally and spatially, highlighting the need for long-term assessments that should be performed separately for different parts of the country.

This paper proceeds as follows: Firstly, we discuss essential aspects and propose basic principles of integrated sustainability assessments of the production of forest raw material in managed forests. Secondly, we outline a conceptual approach for integrated sustainability impact assessments in these environments. Thirdly, we demonstrate our approach through case study examples from two specific counties in Sweden. Finally, we conclude with a discussion and offer suggestions for further research and development in this vital field.

## 2. Materials and Methods

### 2.1. Basic Principles and System Boundaries

This study assesses the sustainability of various forestry practices on the landscape scale across a mosaic of forest stands at various stages of the rotation cycle. Consequently, forest management is guided by a comprehensive, long-term strategy that encompasses all productive forests within the property. This necessitates the assessment of all available productive forest land, including areas designated for purposes other than timber production, rather than solely focusing on the specific stands from which timber is sourced [24,25].

The applied system boundaries in this study for calculating indicators related to forest management practices were defined by the respective county borders and concluded at landings by the roadside. Equivalent indicators for the downstream parts of the forest value chains are not included. As the forest owner holds the legal responsibility for forest management, the focus of the sustainability assessments for forest raw material production should be on the forest owner's forest management decisions [27,28]. Land characterized by sparse forest cover that is unsuitable for profitable forestry, often referred to as an impediment, was not included in the assessment. The resulting assessment values are equally distributed by area across all forest raw materials produced by the forest owner over a specific time frame, in the case of this study a year. Incorporating the assessment values into life-cycle assessments requires them to be correlated with the amounts of forest raw materials generated. However, this introduces further considerations, as well as complexity, and was not addressed in this study.

### 2.2. Choice of Reference Scenarios

In the context of systems analysis of a biological system, a reference scenario must be introduced that typically aims to facilitate the assessment of how the studied "system" influences the aspects of interest [23]. The first ten years of scenario 1: current forest management serve as a reference scenario and constitute the basis for the normalization of the different indicators on a common scale, so that the results can be compared between different indicators, scenarios, and time periods (see Section 2.4 below).

### 2.3. Forest Scenarios

The Heureka forest decision system is a simulation tool, developed by the Swedish University of Agricultural Sciences, for conducting impact analyses and creating action plans within forestry [29,30]. This tool enables analyses using comprehensive data or a selection of data. The time horizon can range from just a few years for tactical planning to over 100 years for harvesting calculations. Heureka PlanWise software is one of three software types in the tool that are applicable on, e.g., stand, estate, landscape, and national level using optimization techniques.

In this study, Heureka PlanWise software, version 2.18.3 was used to calculate different result parameters from forestry, based on simulations from three different scenarios from two counties with large forest cover in Sweden (Figure 1): Västernorrland in the northern part of Sweden (boreal zone) and Kronoberg in the southern part of Sweden (the boreonemoral zone). These two counties differ in terms of natural, climatological, and demographic conditions. Unlike Västernorrland county, a larger portion of the forest in Kronoberg county is owned and managed by individuals and operated as family forestry. Due to its southern location with a milder climate and an extended growth season, Kronoberg county has higher site quality owing to more productive soils [18]. In Kronoberg county, forests cover 81% of the total area, while agriculture accounts for 9%, and built-up areas occupy 8%. Similarly, in Västernorrland, forests cover 85% of the land, agriculture utilizes 2%, and built-up areas represent 3% of the total area.

In this application, Heureka PlanWise modeled the evolution of various attributes of individual forest stands, each representing distinct forested areas. The areas meeting specific criteria were then combined for each of the two counties and compared to the total productive forest areas within each county, as detailed below.

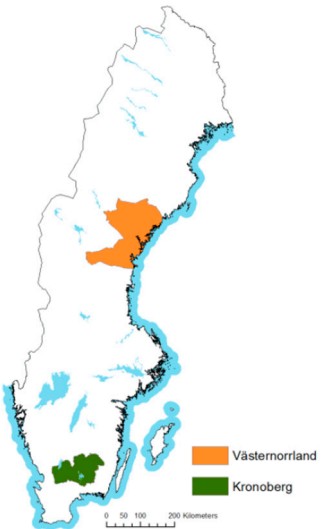

**Figure 1.** A map showing the locations of Kronoberg county in southern Sweden (highlighted in green) and Västernorrland county in northern Sweden (highlighted in orange).

The three scenarios applied in this study offer realistic representations of the potential future trajectory of Swedish forestry. Spanning a century, this study's planning horizon extends from the year 2010 to the year 2110, with outcomes presented in 20 distinct five-year intervals. The scenarios were developed based on different aspects of optimization and should not be seen as a forecast of what will happen in the future. The scenarios used for this study are detailed below.

*Scenario 1: Current forest management*

This scenario reflects a development in which Sweden's forests are used and managed as they are today and as they have been for the past 10–20 years. It describes the development assuming the current direction and level of ambition in forest management and observed harvesting behavior. Harvesting represents potential harvesting and is as high as possible without significantly reducing subsequent harvesting (in the next calculation period), which means it is essentially as high as the net growth (gross growth minus by natural losses) on the part of the productive forest land that is utilized. The first ten years of this scenario serve, as described above, as a reference scenario, the results of which (e.g., estimations of indicator values) can be compared with the results of other scenarios and time periods.

*Scenario 2: Increased forest growth and harvests*

This scenario highlights the potential and effects of increased timber production (growth) given reasonable but high investment levels in forestry. This scenario includes the simulation of several measures to increase production and ensure increased extraction of biomass. These measures are motivated by assumptions of future high demand for timber resources and profitability in forestry. Production enhancement occurs through the optimization and improvement of conventional forestry practices and the introduction of new measures such as the use of the introduced tree species *Pinus contorta* and other exotics/hybrids such as fast-growing tree species, the use of improved plant material, and forest fertilization. In addition, a minimum level of considerations to nature conservation was simulated in the managed parts of the production forest. In this scenario, also, when applicable, the removal of tops and branches is performed in conjunction with both final felling and thinning, along with some stump harvesting during final felling. Additionally, a higher return requirement is imposed by increasing the interest rate in the net present value-maximizing analyses by 1% (from 2% to 3% in Västernorrland county and from 2.5% to 3.5% in Kronoberg county).

*Scenario 3: Double conservation areas*

This scenario reflects a development in which double the areas are set aside, primarily for conservation reasons, compared to Scenario 1. These increased set-asides were simulated in reserves/protected forests, as well as in the enhanced and general nature conservation considerations in connection with forestry actions.

*2.4. Choice of Indicators*

Forestry can be associated with a wide range of sustainability aspects. In this study, the choice was to focus on four important sustainability indicators:

1. Impact on biodiversity;
2. Impact on climate change, separated into fossil and biogenic origin;
3. Impact on the recreational values of the forest;
4. Impact on the economic revenues of the forest owner.

The present study does not encompass additional negative sustainability impacts from forestry such as the delayed or reversed recovery of acidified surface waters [31] and the discharge of nitrogen and mercury into these waters [32,33]. Nonetheless, it is essential to also recognize the potential positive environmental aspects of forestry. These include the role of forests in improving air quality by filtering air pollutants and the purification of runoff water of various pollutants.

*2.5. Indicator Calculations*

2.5.1. Biodiversity

The indicators selected in this study reflect forest stand structures that are regarded as favorable for biodiversity. Such structural indicators are considered useful and cost-effective to use because they are readily visible to non-professionals and can be acquired through forest inventories [34,35]. In our approach, we suggest using four of the biodiversity indicators that are used in the Swedish Environmental Quality Objective 'Sustainable Forests', which are listed below [36]. These four indicators can be regarded as having been established to assess the status of Swedish forests in terms of their structure in order to promote the positive development of biodiversity. Our suggestion is to summarize the areas of forest stands that comply with each of the four indicators and relate these summarized areas to the total areas of productive forest, either in a certain geographical region or the total productive forest under the control of an individual forest owner. In addition, one indicator was included to describe the application of forestry practice with the absence of final fellings, i.e., continuous cover forestry. The five assessed biodiversity indicators used were as follows:

(1) Stands with old trees. The average tree age in the stand should be above 140 years in northern Sweden and above 120 years in southern Sweden;
(2) Stands with dead wood. There should be more than 20 m$^3$ of dead wood per hectare, including only dead wood with a diameter greater than 20 cm;
(3) Stands with large trees. There should be more than 60 trees per hectare with a diameter greater than 45 cm for Norway spruce, Scots pine, and southern broadleaves and a diameter greater than 35 cm for other tree species;
(4) Mixed deciduous and coniferous tree species. The average tree age in the stand should be above 80 years, and more than 3/10 of the basal area should be deciduous tree species;
(5) Stands that are managed with continuous cover forestry, i.e., with no final fellings.

2.5.2. Climate Change (Biogenic)

Annual changes in forest carbon stocks are calculated for living and dead biomass as well as the soil carbon, categorized into mineral and organic (peat) soils. Any climate benefit from temporary carbon storage in forest products ("Harvested Wood Products") was not included in the calculations but arose downstream in the value chain. Substitution

effects were not accounted for either. The used inventory methodology follows, in principle, the methodology used for international climate reporting within land use, land use change, and the forestry sector (LULUCF) [37].

### 2.5.3. Climate Change (Fossil)

The impact of fossil greenhouse gas emissions from forestry on climate change is calculated based on the extent of various forestry management practices (e.g., harvesting, site preparation, etc.), utilizing the life-cycle assessment (LCA)-based emission factors developed by the Forestry Research Institute of Sweden [38]. These emission factors, uniform across the Nordic region, include only fossil greenhouse gas emissions.

### 2.5.4. Social (Recreational Values)

In this study, the impact on the forest's recreational value serves as an indicator of forestry's influence on social values. This was quantified as the cumulative area of forest land meeting criteria for easily accessible forests with good visibility, characterized by sparsely grown, mature productive forests, and forests where the proportion of deciduous trees exceeds 80%. Stands where deciduous trees exceed 80% were considered only if the stand is more than 20 years old. Sparsely grown forests were defined as areas with a maximum of 1000 tree stems per hectare and a stand age exceeding 50 years. Furthermore, areas with productive forest stands that are devoid of final-cut harvests over the past ten years were included as the fraction exceeding 80%. Finally, areas where the recreation index value, calculated by Heureka, exceeded a threshold of 0.6 were also included. The Heureka recreation index values were calculated based on [39]. The sum of the areas meeting these criteria is related to the total area of productive forest land as a fraction from 0 to 1. Due to the possibility of forest areas meeting multiple criteria, double accounting may occur, leading to rare instances where the final recreation index value may exceed 1.0.

### 2.5.5. Economic Revenues

The economic outcome of forestry is of significant importance for individual forest owners. The net economic revenue for the forest owner is calculated by Heureka PlanWise, based on forest owners' incomes and costs.

### 2.6. Normalization to a Common Scale

To translate absolute values for a specific sustainability indicator to a common scale, reference scenarios were employed alongside preference and indifference values (Figure 2). The characterization factor, CF, is the normalized value on a common scale and calculated separately for each sustainability indicator. In calculations pertaining to an individual forest owner, a zero value on the common, relative scale was derived from a reference scenario based on the starting point of the current average forest conditions as of 2010 for the specific aspect of sustainability at the regional level [39,40]. In this study, where calculations were conducted only at the county level, the state of the "Current forest management" scenario over the initial ten years served as the reference scenario, as described above.

Preference and indifference values are used to delineate the numerical values on the relative scale. Preference values represent a level of the absolute value of the indicator deemed "good enough", drawing from various target values for environmental quality objectives or similar criteria. Indifference values signify a threshold below which a change in the absolute value of the indicator is no longer deemed significant, akin to "it doesn't get worse". The calculations of normalized characterization factors reflecting the relative importance from absolute numerical values for various sustainability indicators are illustrated in Figure 2. A positive value on the normalized scale represents a positive impact from a sustainability perspective and vice versa.

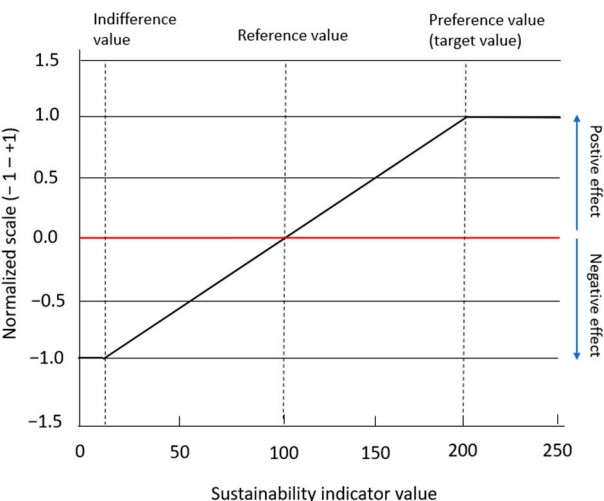

**Figure 2.** Illustration of the methodology used to transfer absolute values for a specific sustainability indicator value (x-axis, for this example an arbitrary value) to normalized characterization factors on a relative scale between −1 and +1 (y-axis), which can be used for comparisons of different sustainability indicators. A positive value on the normalized scale represents a positive impact from a sustainability perspective and vice versa. The characterization factor, CF, is the normalized value on a common scale and calculated separately for each sustainability indicator. Adapted from [40].

In the final application for the future, preference, and indifference values are to be based on different target values used in environmental target monitoring or, alternatively, on scientific assessments of the levels considered desirable to achieve. In this pilot study, a simplified relationship between preference and indifference values was implemented by using a reference value to set the other two values, which were used for educational purposes to facilitate the understanding of how the different indicators function in relation to results from Heureka PlanWise. In Table 1, the values used in this study as reference, preference, and indifference values for Kronoberg and Västernorrland counties are provided. Note that there is often no symmetry when the preference and indifference value are defined, and, therefore, the relative order between impact category results is different compared to what would be observed if a simple normalization was used to calculate results.

**Table 1.** Reference, preference, and indifference values used for different indicators for the counties of Kronoberg and Västernorrland. The value for Kronoberg is given before the slash, while the value for Västernorrland is given after the slash. Reference scenario is scenario 1: current forest management, covering period 1–10 years for all indicators.

| Indicator | Unit | Reference Value | Preference Value | Indifference Value | Preference | Indifference |
|---|---|---|---|---|---|---|
| Biodiversity | Fraction of the total area of productive forest | 0.29/0.31 | 0.57/0.63 | 0.029/0.031 | Double reference value | 10% of reference value |
| Climate (biogenic) | Tonnes $CO_{2e}$ $ha^{-1}$ $yr^{-1}$ | 3.27/0.37 | 6.54/3.27 | 0.33/0.164 | Kronoberg: Double reference value. Västernorrland: 50% of the Kronoberg preference value * | Kronoberg: 10% of reference value. Västernorrland: 50% of the Kronoberg preference value * |
| Climate (fossil) | Tonnes $CO_{2e}$ $ha^{-1}$ $yr^{-1}$ | 0.0135/0.016 | 0.0013/0.0016 | 0.0270/0.031 | 10% of reference value | Double reference value |
| Social (recreation) | Fraction of the total area productive forest | 0.43/0.35 | 0.87/0.70 | 0.043/0.035 | Double reference value | 10% of reference value |
| Economic (net income) | SEK $ha^{-1}$ $yr^{-1}$ | 613/772 | 1227/1543 | 61/77 | Double reference value | 10% of reference value |

* Since the reference value for Västernorrland was close to zero, it was not useful to assume preference values to be twice the reference value and the indifference value as 10% of the reference value. Instead, 50% of the preference and indifference value from Kronoberg was used. The use of 50% was motivated by the far northern location of Västernorrland, compared to Kronoberg county.

### 2.7. Integration and Visualization

The last step included the visualization of results. Here, spider diagrams were used to conceptualize and communicate the results from this study.

## 3. Results and Discussion

### 3.1. Timber Production

Sustainability assessments for forestry applied in the three different scenarios need to be related to the amount of forest resources generated in each county. The production of forest resources is reported as a total for each county and across all tree species and assortments, based on the sum of the result variables from Heureka PlanWise, and expressed in the unit m³ fub year⁻¹ (Figure 3). This represents the annual cumulative extraction of sawlogs and pulpwood from the county. The production of energy wood, logging residues, and possible stump harvest are not included. The average yearly production of forest raw materials is shown for two periods, 10–50 and 50–100 years, for each scenario, summed up for the entire county (Table 2). The production of timber per hectare was almost twice as high in the county of Kronoberg compared to Västernorrland (Table 2). This is mainly caused by the differences in climate between the two counties, with a colder climate and a shorter growing season in Västernorrland.

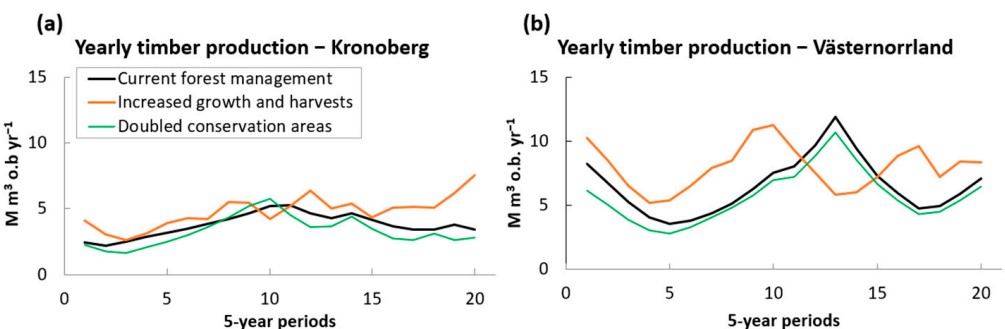

**Figure 3.** Yearly timber production estimated for 5-year periods over 100 years under three different forestry scenarios for Kronoberg county (**a**) and Västernorrland county (**b**). The estimates include production for all tree species and all timber qualities.

**Table 2.** The total production of forest raw materials summarized over two different periods, 10–50 years and 50–100 years, for each scenario, summed up for the entire county. The unit is m³fub ha⁻¹ yr⁻¹. The calculations include the combined production of timber and pulpwood from all tree species but do not include the production of energy wood, logging residue, small trees, and stumps.

| | Current Forestry | Increased Growth and Harvests | Doubled Conservation Areas |
|---|---|---|---|
| Kronoberg, 10–50 years | 0.99 | 1.10 | 0.93 |
| Kronoberg, 50–100 years | 1.08 | 1.47 | 0.89 |
| Västernorrland, 10–50 years | 0.49 | 0.76 | 0.42 |
| Västernorrland, 50–100 years | 0.73 | 0.77 | 0.67 |

The yearly production of forest raw materials varied over time depending on both the initial age structure of the forest stands and the applied forest management method. At the beginning of the time series (years 2000–2010), the stands in Kronoberg county were relatively young, resulting in lower production, which generally increased over the next fifty years (Figure 3a). The initial production was highest in the "increased growth and harvests" scenario due to the replacement of some stands for more productive tree species. In contrast, production was lower in the "doubled conservation areas" scenario due to

increased set-aside conservation areas. Over the next fifty years, timber production was notably highest in the "increased growth and harvests" scenario, followed by "current forest management", and lowest in the "doubled conservation areas" scenario. As mean values for the period 10–50 years and compared to "current forest management" scenario, production was 11% higher in the "increased growth and harvests" scenario and 6% lower in the "doubled conservation areas" scenario (Table 2). The corresponding values for the 50–100-year period were 36% higher in the "increased growth and harvests" scenario and 18% lower in the "doubled conservation areas" scenario.

In Västernorrland county, the forest stands were initially older, leading to initially higher annual production across all scenarios. However, this production subsequently declined during the first 25 years in all scenarios (Figure 3b). Subsequently, there were differences in the development over time for production, with faster growth observed in the "increased growth and harvests" scenario, peaking after around 50 years, while the maximum was reached later in the other two scenarios. The production was relatively similar between the "current forest management" and the "doubled conservation areas" scenarios. As mean values for the period 10–50 years and compared to the "current forest management" scenario, production was 56% higher in the "increased growth and harvests" scenario and 13% lower in the "doubled conservation areas" scenario (Table 2). The corresponding values for the 50–100 years period were 5% higher in the "increased growth and harvests" scenario and 9% lower in the "doubled conservation areas" scenario.

To summarize, production in the "increased growth and harvests" scenario was substantially increased compared to the "current forest management" scenario, but the time scales differed between the two counties. In contrast, production was reduced in the "doubled conservation areas" scenarios, but the difference compared to "current forest management" was not as pronounced.

### 3.2. Impact on Biodiversity

Biodiversity indicator values ($BD_i$) and characterization factors ($CF_{BD}$) for impacts from the different forestry scenarios on biodiversity are shown in Figure 4. The $BD_i$ increases over time in both counties for both the "current forest management" and "doubled conservation areas" scenarios, while it remains relatively constant in the "increased growth and harvests" scenario. In both Kronoberg and Västernorrland, the $BD_i$ reached the preference value at around years 50–60 in the scenario "doubled conservation areas". In the "current forest management" scenario, the preference value was reached towards the end of the 100-year period in Kronoberg, but it was not achieved in Västernorrland. In all scenarios, the most influential factor in the $BD_i$ calculation was the areas with deadwood trees, followed by the areas with old forests.

Comparing mean values for the period 10–50 years in Kronoberg county to the "current forest management" scenario, the $BD_i$ was 42% lower in the "increased growth and harvests" scenario and 20% higher in the "doubled conservation areas" scenario (Table 3). The values for the 50–100-year period were 48% lower in the "increased growth and harvests" scenario and 36% higher in the "doubled conservation areas" scenario. The corresponding values for Västernorrland county, for the period of 10–50 years and compared to the "current forest management" scenario, were 23% lower in the "increased growth and harvests" scenario and 22% higher in the "doubled conservation areas" scenario (Table 3). The values for the 50–100-year period were 50% lower in the "increased growth and harvests" scenario and 37% higher in the "doubled conservation areas" scenario.

To summarize, the indicator values for biodiversity increased over time in both the "current forest management" scenario and the "doubled conservation areas" scenario in both counties, compared to the reference situation, which was the first ten years of the "current forest management" scenario. The increase was most substantial in the "doubled conservation areas" scenario. In the "increased growth and harvests" scenario, the biodiversity indicator values remained at approximately the same level as the reference situation. The development of the biodiversity indicator value would to a large extent

depend on the stand age, which, in turn, depends on the initial stand age distribution, as well as the frequency of clearcut harvests.

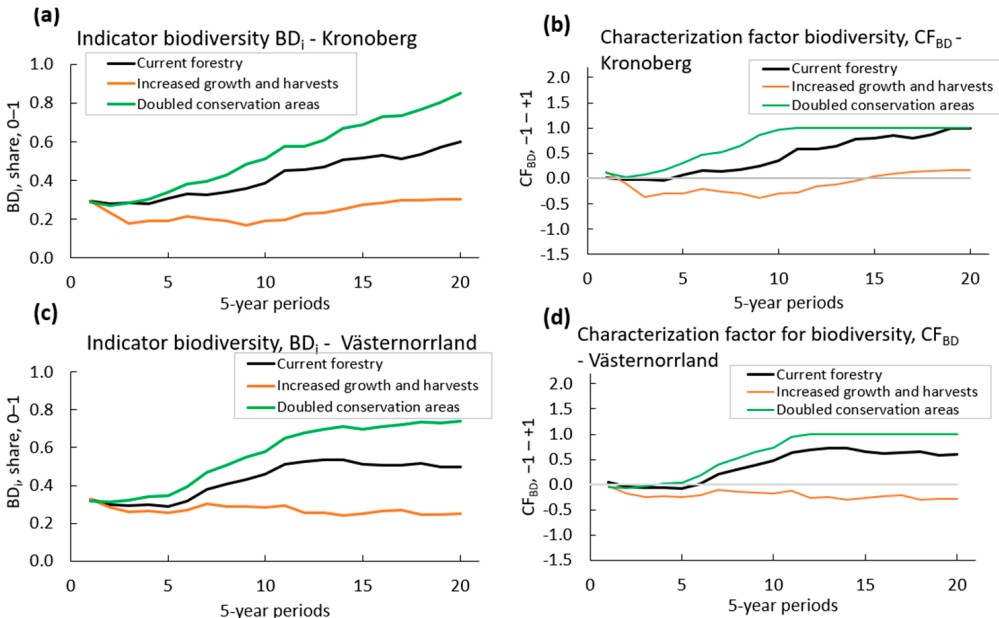

**Figure 4.** Biodiversity indicator values ($BD_i$, **a**,**c**) and characterization factors ($CF_{BD}$, **b**,**d**) for impacts from three different forestry scenarios on biodiversity over 100 years in Kronoberg county (**a**,**b**) and Västernorrland county (**c**,**d**). $_{BDi}$, biodiversity index; $CF_{BD}$, characterization value for biodiversity.

**Table 3.** Calculated biodiversity indicator values, as the average for two different periods, 10–50 years and 50–100 years of the 25-year periods, spanning 100 years, for each scenario and county. The unit is in the range of 0–1. Due to potential double counting, the calculated value may exceed 1.0. The calculations are based on the entire productive forest land area within the county.

|  | Current Forestry | Increased Growth and Harvests | Doubled Conservation Areas |
|---|---|---|---|
| Kronoberg, 10–50 years | 0.33 | 0.19 | 0.39 |
| Kronoberg, 50–100 years | 0.52 | 0.27 | 0.70 |
| Västernorrland, 10–50 years | 0.36 | 0.28 | 0.44 |
| Västernorrland, 50–100 years | 0.52 | 0.26 | 0.71 |

### 3.3. Impact on Climate Change (Biogenic)

Biogenic climate change impact indicator values ($CL_{bio}$) and characterization factors ($CF_{CLbio}$) are shown in Figure 5. Initially, in both Kronoberg and Västernorrland counties, the $CL_{bio}$ showed more carbon being stored (sequestered) in the forest. In Kronoberg, this was primarily due to the younger age of forest stands, leading to less frequent harvesting.

A key factor affecting the $CL_{bio}$ across all scenarios was the change in the living biomass carbon stocks. This aligns with the findings outlined in the Swedish National Inventory Report to the Climate Convention [41].

In both Kronoberg and Västernorrland, the $CL_{bio}$ values increased over time (changed in a positive direction) and reached the indifference values at around years 40–50 in all scenarios, resulting in the $CF_{CLbio}$ approaching −1.0. The indifference values were the very low values for carbon sequestration, being 0.33 and 0.16 tonnes $CO_{2e}$ ha$^{-1}$ yr$^{-1}$, respectively (Table 1). This reflected the fact that the forest ecosystems, for some time, became a source for $CO_2$ to the atmosphere. During the latter stages of the 100-year period, the forest ecosystems again became carbon sinks. However, in Kronoberg, only the "doubled conservation areas" scenario managed to reach the reference value for carbon sequestration.

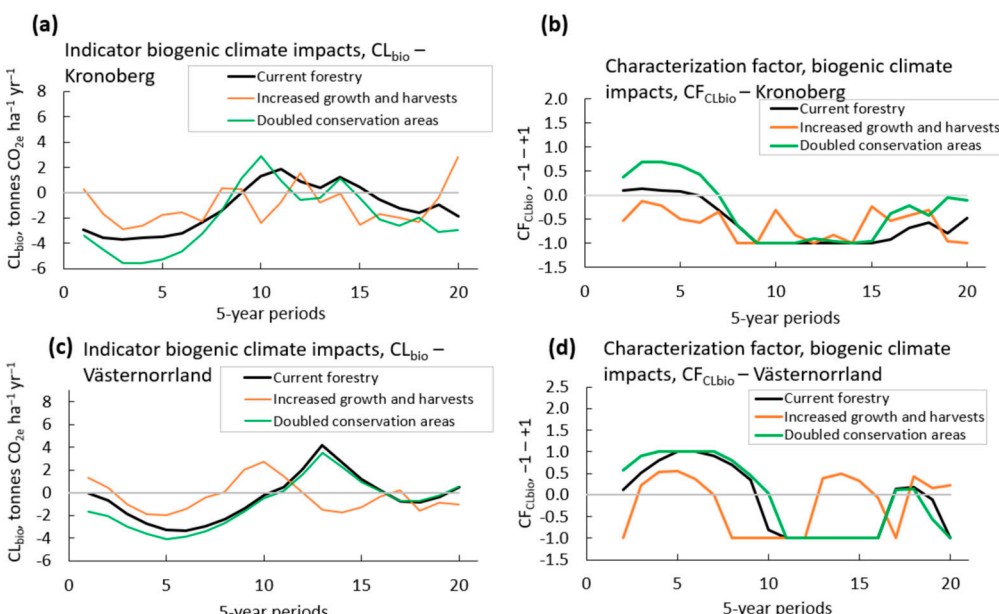

**Figure 5.** Indicator values (CL$_{bio}$, **a**,**c**) and characterization factors (CF$_{CLbio}$, **b**,**d**) from four different forestry scenarios on biogenic climate impacts over 100 years in Kronoberg county (**a**,**b**) and Västernorrland county (**c**,**d**). CL$_{bio}$, the biogenic climate impacts index; CF$_{Clbio}$, characterization value for biogenic climate impacts. Since the reference value for Västernorrland was close to zero, it was not useful to assume the preference value to be twice the reference value and the indifference value to be 10% of the reference value. Instead, 50% of the preference and indifference value from Kronoberg was used. The use of 50% was motivated by the far northern location of Västernorrland compared to Kronoberg county. The reference, preference, and indifference values used for Kronoberg were 3.3, 6.5, and 0.33 tonnes $CO_{2e}$ ha$^{-1}$ yr$^{-1}$ and for Västernorrland were 0.37, 3.3, and 0.16 tonnes $CO_{2e}$ ha$^{-1}$ yr$^{-1}$.

When comparing the average CL$_{bio}$ values over 10–50 years in Kronoberg with the "current forest management" scenario, the "increased growth and harvests" scenario showed a 22% reduction, while the "doubled conservation areas" scenario showed a 32% increase (Table 4). Over 50–100 years, the "doubled conservation areas" scenario had the highest carbon sink, followed by the "increased growth and harvests" scenario, while the "current forest management" scenario had the lowest carbon sink. For the period of 10–50 years in Västernorrland county, when compared to the "current forest management" scenario, the carbon sink was reduced by 89% in the "increased growth and harvests" scenario and was 26% higher in the "doubled conservation areas" scenario (Table 4). In the subsequent 50-year period, there was a small carbon sink in the "increased growth and harvests" scenario, while both the "current forest management" and the "doubled conservation areas" scenarios became sources of $CO_2$ to the atmosphere.

The results for the biogenic climate change impact indicator values illustrate the consequences of the initial state of the forests at the time of the start of the analysed hundred-year period, as well as the long time lag between changes in forest management and the subsequent changes in the forest carbon stocks. In the county of Kronoberg, the forests were relatively young, mainly as a result of replanting trees after major windthrow caused by a severe storm in southern Sweden in 2005 [42]. As a result, the frequency of clearcut harvests was relatively low during the first 30 years, and the living biomass carbon stocks increased considerably. Subsequently, the clearcut harvests gradually increased and due to the forests' relatively uniform age structure, many forests were clearcut harvested after approximately 50 years. Consequently, the forests in Kronoberg became a source for $CO_2$ emissions to the atmosphere for several years. The different forest management scenarios did not substantially affect these changes in the forest carbon stocks over time. In the scenario "increased growth and harvests", more forest stands were initially clearcut harvested in

order to be replaced with more productive tree varieties and species. Subsequently, the frequency of clearcut harvests was more evenly distributed, and the forests remained a small sink for a majority of the time. In the "doubled conservation areas" scenario, clearcut harvests were initially kept at a lower clearcut harvest frequency, but eventually many of these stands had to be clearcut harvested, and these forests then became a source for $CO_2$ emissions to the atmosphere for a few years. In Västernorrland county, the forest stands were, on average, older during the first part of the period, and the frequency of clearcut harvests was relatively high. Therefore, these forests were initially close to carbon-neutral. However, the carbon stock changes over time in Västernorrland for the different scenarios were rather similar to those observed for Kronoberg.

**Table 4.** Calculated absolute values for the indicator of biogenic climate impact as the sum of values for two different periods, 10–50 years and 50–100 years of the 25-year periods, spanning 100 years, for each scenario and county. The unit is tonnes of $CO_{2e}$ per hectare per year. A negative value indicates the removal of $CO_2$ from the atmosphere to the forest. The calculations are based on the entire productive forest land area within the county.

| | Current Forestry | Increased Growth and Harvests | Double Conservation Areas |
|---|---|---|---|
| Kronoberg, 10–50 years | −2.06 | −1.61 | −2.72 |
| Kronoberg, 50–100 years | −0.15 | −0.64 | −1.22 |
| Västernorrland, 10–50 years | −2.27 | −0.25 | −2.86 |
| Västernorrland, 50–100 years | 0.92 | −0.67 | 0.74 |

*3.4. Impact on Climate Change (Fossil)*

Fossil climate change impact indicator ($CL_{fos}$) values and their corresponding characterization factors ($CF_{cLfos}$) are shown in Figure 6. $CL_{fos}$ values were estimated based on $CO_2$ emission factors coupled to different forest management practices. Hence, $CL_{fos}$ is strongly coupled to the production of forest raw materials. Furthermore, it should be noted that the values for $CL_{fos}$ are approximately 100 times lower than those for $CL_{bio}$ (see Figure 5 for comparison). As expected, $CL_{fos}$ was consistently estimated as emissions of $CO_2$ to the atmosphere, with the highest values observed in the "increased growth and harvests" scenario and the lowest in the "doubled conservation areas" scenario. Since the production generally increased over time, the $CL_{fos}$ reached the indifference values in all scenarios and both counties. However, in Kronoberg county, the indifference value was exceeded for the longest duration in the "increased growth and harvests" scenario.

Comparing mean values for the period of 10–50 years in Kronoberg, we referred back to Table 1 from where it first mentioned. For the "current forest management" scenario, the $CL_{fos}$ values were 23% higher in the "increased growth and harvests" scenario and 7% lower in the "doubled conservation areas" scenario (Table 5). The values for the 50–100-year period were 53% higher in the "increased growth and harvests" scenario and 17% lower in the "doubled conservation areas" scenario. The corresponding values for Västernorrland county were for the period of 10–50 years and compared to the "current forest management" scenario, being 55% higher in the "increased growth and harvests" scenario and 12% lower in the "doubled conservation areas" scenario (Table 5). The values for the 50–100-year period were 17% higher in the "increased growth and harvests" scenario and 10% lower in the "doubled conservation areas" scenario.

To conclude, the fossil emissions of $CO_2$ were tightly coupled to the forest management activities and, as expected, highest in the "increased growth and harvests" scenario and lowest in the "doubled conservation areas" scenario.

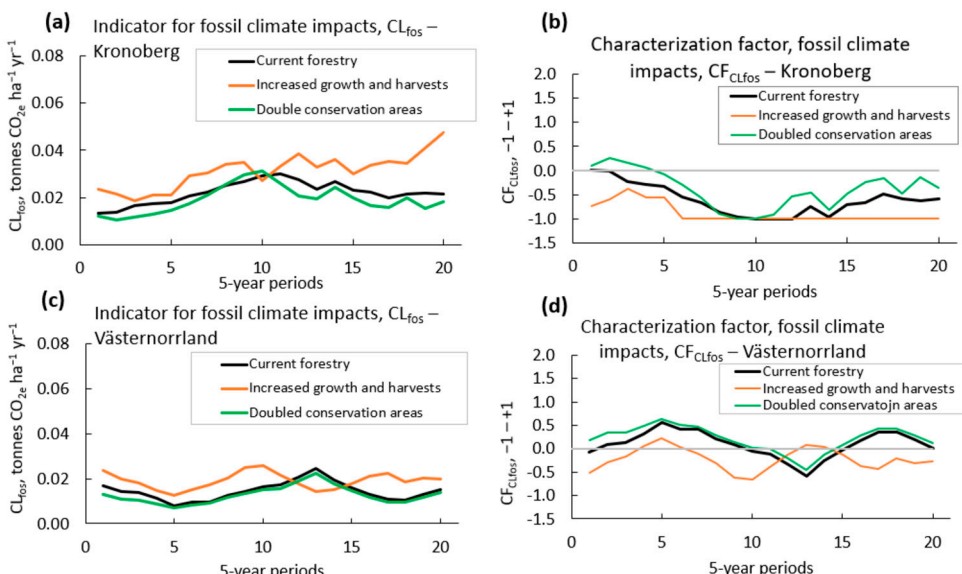

**Figure 6.** Indicator values (CL$_{fos}$, **a**,**c**) and characterization factors (CF$_{cLfos}$, **b**,**d**) for impacts from four different forestry scenarios on fossil GHG emissions over 100 years in Kronoberg county (**a**,**b**) and Västernorrland county (**c**,**d**). CL$_{fos}$, the fossil climate impacts index; CF$_{cLfos}$, characterization value for fossil climate impacts. Note that a negative characterization value represents a positive impact from a sustainability point of view and vice versa.

**Table 5.** Calculated values for the indicator of climate impact from emissions of fossil greenhouse gases related to forestry as the sum of values over all twenty five-year periods, spanning 100 years, for each scenario and county. Unit in tonnes of $CO_2$e per hectare per year. The area used for calculations is the county's total area of productive forest land. A positive value indicates a release of $CO_2$ into the atmosphere.

|  | Current Forestry | Increased Growth and Harvests | Double Conservation Areas |
|---|---|---|---|
| Kronoberg, 10–50 years | 0.022 | 0.027 | 0.020 |
| Kronoberg, 50–100 years | 0.024 | 0.036 | 0.020 |
| Västernorrland, 10–50 years | 0.012 | 0.019 | 0.011 |
| Västernorrland, 50–100 years | 0.016 | 0.019 | 0.015 |

### 3.5. Impacts on Recreation Values

Recreation value impact indicator (SO$_{recr}$) values and the corresponding characterization factors (CF$_{SOrecr}$) are shown in Figure 7. In both counties, the SO$_{recr}$ increased over time, compared to the reference value (which was the value for the first ten years in the "current forest management" scenario), in the "current forest management" and "doubled conservation areas" scenarios. The CF$_{SOrecr}$ varied above and below zero in the "increased growth and harvests" scenario.

As mean values for Kronoberg county for the period of 10–50 years and compared to the "current forest management" scenario, the SO$_{recr}$ values were 46% lower in the "increased growth and harvests" scenario and 16% higher in the "doubled conservation areas" scenario (Table 6). The values for the 50–100-year period were 35% lower in the "increased growth and harvests" scenario and 12% higher in the "doubled conservation areas" scenario. The corresponding values for Västernorrland county were for the period of 10–50 years and compared to "current forest management" scenario, being 36% lower in the "increased growth and harvests" scenario and 11% higher in the "doubled conservation areas" scenario (Table 6). The values for the 50–100-year period were 27% lower in the

"increased growth and harvests" scenario and 16% higher in the "doubled conservation areas" scenario.

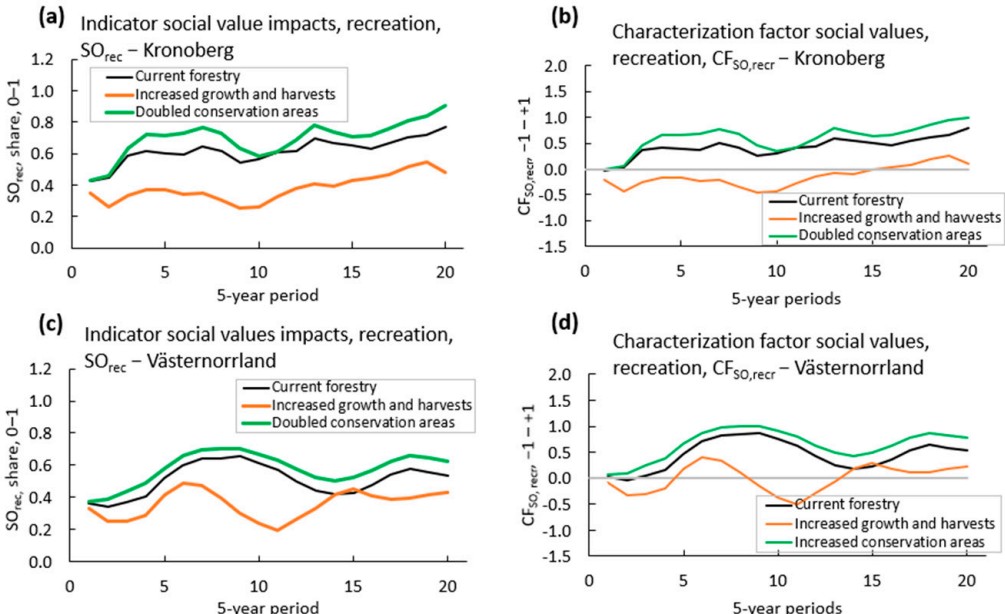

**Figure 7.** Indicator values (**a**,**c**) and characterization factors (**b**,**d**) for impacts from three different forestry scenarios on recreation values over 100 years in Kronoberg county (**a**,**b**) and Västernorrland county (**c**,**d**). $SO_{rec}$, social value impacts for the recreation index; $CF_{sOrec}$, characterization value for social values for the recreation impacts.

**Table 6.** Calculated values for the indicator of impact on the recreational value of the forest in connection with forestry, as averages for two different periods, 10–50 years and 50–100 years, of the 25-year periods, spanning 100 years, for each scenario and county. The unit is in the fraction range of 0–1. The area used for calculations is the county's total area of productive forest land.

|  | Current Forestry | Increased Growth and Harvests | Double Conservation Areas |
| --- | --- | --- | --- |
| Kronoberg, 10–50 years | 0.60 | 0.32 | 0.69 |
| Kronoberg, 50–100 years | 0.67 | 0.44 | 0.76 |
| Västernorrland, 10–50 years | 0.56 | 0.36 | 0.62 |
| Västernorrland, 50–100 years | 0.50 | 0.37 | 0.59 |

In conclusion, the recreation value increased over time in both the "current forest management" and "doubled conservation areas" scenarios, while it remained relatively constant in the "increased growth and harvests" scenario. These patterns were relatively similar to the changes over time in the biodiversity indicator (Figure 3), since the criteria used for the definitions of favorable forest structures for biodiversity and recreation values were to some extent similar.

### 3.6. Impacts on Economic Values

Economic value impact indicator values ($EK_{netrev}$) and the corresponding characterization factors ($CF_{EKnetrev}$) are shown in Figure 8. It is evident that the mean economic revenue for the forest owners is closely coupled to timber production (Figure 3).

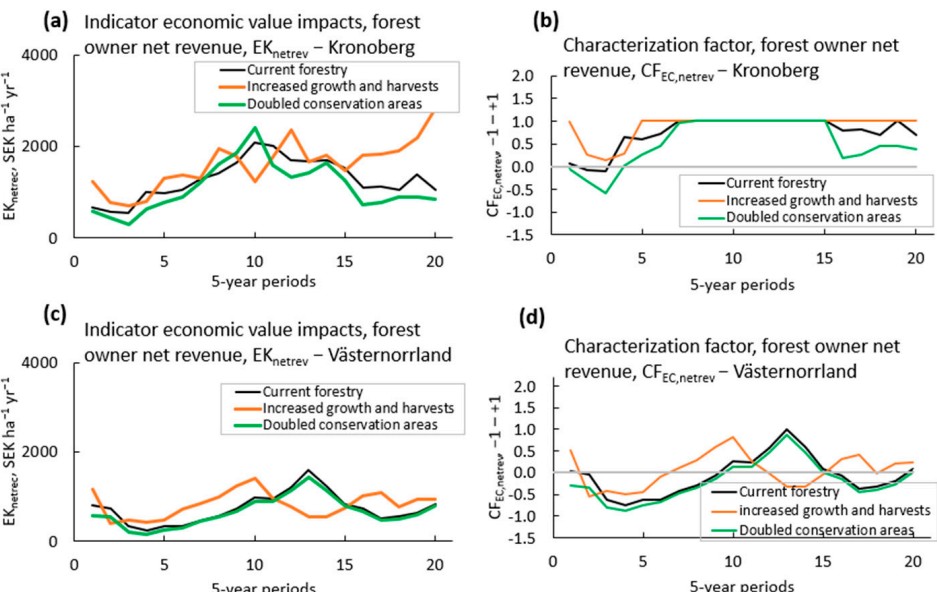

**Figure 8.** Indicator values (**a**,**c**) and characterization factors (**b**,**d**) for impacts from four different forestry scenarios on economic values, expressed as the yearly net revenue for the forest owner over 100 years in Kronoberg county (**a**,**b**) and Västernorrland county (**c**,**d**). $EC_{netrev}$, economic values' impact on forest owner yearly net revenue; $CF_{EC, netrev}$, characterization values for economic values impacts on forest owner yearly net revenue.

The $EK_{netrev}$ increased over time in the first part of the 100-year period in all scenarios in Kronoberg county, and the preference value was reached during large parts of the period. In Västernorrland county, the $EK_{netrev}$ varied over time but were relatively similar between the "current forest management" and "doubled conservation areas" scenarios.

As mean values for Kronoberg county for the period of 10–50 years and compared to the "current forest management" scenario, the $EK_{netrev}$ were 4% higher in the "increased growth and harvests" scenario and 4% lower in the "doubled conservation areas" scenario (Table 7), i.e., relatively small changes. The values for the 50–100-year period were 37% higher in the "increased growth and harvests" scenario and 20% lower in the "doubled conservation areas" scenario. The corresponding values for Västernorrland county were for the period of 10–50 years and compared to the "current forest management" scenario, being 65% higher in the "increased growth and harvests" scenario and 13% lower in the "doubled conservation areas" scenario (Table 7). The values for the 50–100-year period were 8% lower in both scenarios.

To summarize, the economic revenue of the forest owners in Kronoberg county increased considerably over time, with highest values recorded in the "increased growth and harvests" scenario. The difference between the "current forest management" and the "doubled conservation areas" scenarios was relatively small. This similarity was also observed in Västernorrland county. The economic revenue of the forest owners per hectare was lower in Västernorrland, mostly due to the lower forest growth rates in this northern climate.

**Table 7.** Calculated values for the indicator of the forest owner's annual net income per unit of productive forest land as averages for two different periods, 10–50 years and 50–100 years, of the 25-year periods spanning 100 years, for each scenario and county. The unit is SEK ha$^{-1}$ year$^{-1}$. The area used for the calculations is the county's total area of productive forest land.

|  | Current Forestry | Increased Growth and Harvests | Double Conservation Areas |
|---|---|---|---|
| Kronoberg, 10–50 years | 1251 | 1303 | 1206 |
| Kronoberg, 50–100 years | 1428 | 1957 | 1137 |

**Table 7.** *Cont.*

|  | Current Forestry | Increased Growth and Harvests | Double Conservation Areas |
|---|---|---|---|
| Västernorrland, 10–50 years | 497 | 818 | 430 |
| Västernorrland, 50–100 years | 906 | 830 | 835 |

### 3.7. Integration of Results for Different Indicators

The mean values for the characterization factors (CF) for the five different indicators and the two counties are shown in the spider diagrams in Figure 9, being for the two different time periods of 10–50 years and 50–100 years.

The absolute values of sustainability indicators were transferred to relative characterization factor values between −1 and +1, as described above. The reference scenario value of zero was represented by the middle circle in the diagrams. The outer circle represented the characterization factor value of +1, which represents the most positive aspect of the indicator from a sustainability point of view, based on the applied preference values. The midpoint value in each diagram and indicator represents the worst case from a sustainability standpoint (i.e., the CF value of −1), calculated based on the indifference values assigned to each indicator.

In Kronoberg county, the mean CF values during the 10–50-year period for the "current forestry scenario" became worse from a sustainability point of view (i.e., closer to the midpoint of the spider diagram in comparison with the reference value) for the biogenic $CO_2$ and fossil $CO_2$. However, there was an improvement for the biodiversity, recreation, and owner's economic net revenue CF. This pattern remained for the 50–100-year period. Also, for the "increased growth and harvest" scenario", the biogenic $CO_2$ and fossil $CO_2$ CF became worse during the entire 100-year period, while the CF for biodiversity and recreation first became worse but for the 50–100 year returned close to the reference values (i.e., the middle circle). The owner's net revenue CF was, in this scenario, high during the entire 100-year period. In the "double conservation area" scenario, the CF for biodiversity improved during the entire 100-year period, while the biogenic and fossil $CO_2$ CF became worse, especially during the 50–100-year period. The recreation value and owner's net revenue both improved during the entire 100-year period.

The pattern of changes was relatively similar for Västernorrland county in terms of CF for biodiversity and recreation. The CF for biogenic $CO_2$ improved for the "current forestry" and "doubled conservation areas" scenarios during years 10–50 but decreased during years 50–100, while it did not change much for the "increased growth and harvest" scenario. The fossil $CO_2$ CF was improved or remained unchanged, compared to the reference value, in all scenarios during the entire 100-year period. The owner's net revenue CF became worse or remained unchanged for all scenarios during the entire 100-year period. Overall, in Västernorrland, the CF did not change much for the "increased growth and harvest" scenario for any of the two periods.

Overall, the forestry practices in the scenario "doubled conservation areas" resulted in the most positive CF values from a sustainability point of view for most of the different indicators. Conversely, the scenario of "increased growth and harvests" produced the least favorable CF values, being closest to the midpoint of the spider diagrams. As expected, the owner's net revenue was the exception, where the scenario "increased growth and harvests" generated the most positive CF values, even though the differences between the scenarios were not substantial. It should be kept in mind that the CF values for the different indicators were calculated relative to a reference value, which was the value for the respective indicator for the first ten years in the "current forestry scenario". Hence, the comparisons of changes over time will be in relation to the current situation for the forests in the respective counties.

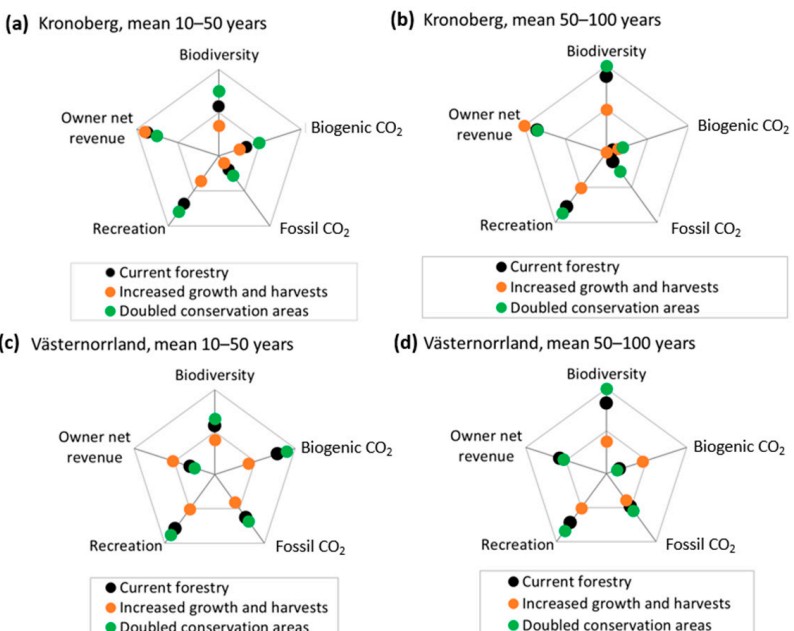

**Figure 9.** Spider diagrams illustrating characterization factors for five different sustainability indicators estimated for three different future forestry scenarios för the counties of Kronoberg (**a,b**) and Västernorrland (**c,d**). The different forestry scenarios were calculated over a 100-year period, starting in 2010.

## 4. Discussion

The calculations of sustainability indicators for the three different scenarios of future forestry over a 100-year period have yielded expected results. The significance of the time frame has also become evident in the calculations of characterization factors based on assumed values for the reference, preference, and indifference scenarios.

The three scenarios analyzed in detail, one of which represents "Current forest management", are considered entirely realistic. There were differences between the two counties concerning calculated indicators for the scenario "Current forest management". Over the 100-year period, the production of forest raw materials per forest area of productive forests was 70% higher in Kronoberg county compared to Västernorrland county (Table 2). The indicator value for biodiversity was similar between the two counties, while the indicator for recreational value was 20% higher in Kronoberg. Indicators for both biogenic and fossil climate impact were approximately 60% higher in Kronoberg county, and the economic indicators were 90% higher. All these differences can be explained by the more southern location of Kronoberg county.

As expected, the calculated indicator values for the "Increased growth and harvests" scenario show a higher occurrence of negative values, from a sustainability point of view and compared to other scenarios, for all indicators, except forest owner net economic revenue. However, concerning biogenic climate benefit, the "Increased growth and harvests" scenario provided more positive values in the later part of the 100-year period for Västernorrland county.

The calculated values in the "Double Nature Conservation Areas" scenario resulted in more favorable impacts on most indicators compared to the scenarios "Current Forestry Practices" and "Increased Production, except for forest owner net economic revenue, where the reverse was the case".

Overall, the results indicate the importance of testing how the calculated values for various indicators behave in forestry across different parts of the country. Furthermore, it is crucial to assess an individual forest owner based on the geographical, climatic, and environmental conditions prevailing in their own county, as is the case in the proposed

methodology [39,40]. It is also essential that assessments are performed over long time periods, as the impact on the various indicators varies over time.

An important aspect regarding the selection of indicators is that it should be possible to obtain quantitative values with a reasonable amount of effort. The ongoing digitalization transforming the forestry sector can facilitate this development, emphasizing the importance of digitizing and storing sustainability indicator data in an accessible and transparent manner. Traceability also forms the basis of sustainability calculations to produce a specific forest commodity or product. The rapid advance of artificial intelligence and the use of high-resolution remote sensing to detect, map, and monitor spatio-temporal changes in forests in Sweden and other parts of the world will provide detailed information about forest structures and conditions at lower costs. Information can also be obtained from the forest owner's own inventories. Regarding forestry actions, such as harvesting activities, input data will become available through semi-automatic and remote-controlled forestry machines. It is also desirable for proposed indicators to be linked to information already generated by various forest owners, such as that used for certification under FSC/PEFC standards.

The calculations made in this study have been restricted to apply within the borders of Sweden and are at this stage only applicable for forestry practices. Sustainability assessments for the downstream value chain will require more research, as some indicators reflecting the sustainability of industrial and manufacturing processes or a wood-based product are different [43]. Carbon storage in wood products for a certain time period and substitution effects are not yet included in our analysis. However, incorporating these factors could enhance our understanding of changes in forest management practices and resulting changes in the harvest compositions [44]. The complexity of the calculations also increases significantly if wood exports and imports are considered. Regarding exports, uncertainties arise regarding how the products are ultimately used. Concerning imports, uncertainties arise regarding the circumstances under which forest resources and products have been produced.

The implementations of our framework could, however, be adapted to other Nordic countries, as there are many similarities and characteristics between these countries in terms of forest and forestry [45]. For example, since the 1990s, the Nordic Forestry Model has been implemented, aiming to provide a competitive, highly productive forestry industry, as well as ensure the preservation of biodiversity in forest production landscapes through collaborative endeavors by both the government and forest owners [46]. There are also similarities in ownership as private individuals and families own about half of the forest land, and as such, individual forest owners in the Nordic countries have a relatively large degree of freedom regarding which forest management goals to apply [47,48].

## 5. Conclusions and Future Research

In this study, we have presented a framework for integrated sustainability impact assessments using widely different indicators for managed forests in Sweden. The framework is flexible and can be applied at various geographical and temporal scales. The findings show the significance of assessing indicator performance in forestry across various geographical regions and customizing assessments for individual forest owners based on the geographical, climatic, and environmental conditions prevailing in their own region. Additionally, long-term assessments are essential due to the varying impacts of indicators over time, to a large extend depending on the forest structure at the time of the start of the assessment period. Applications of the framework include describing the consequences of alternative future scenarios for the production and use of forest raw materials in Sweden. The method can also be used as a basis for disclosing the environmental performance of a product based on forest raw materials, for example, within life-cycle analysis or sustainability reporting. The methodology suggested here for calculating various sustainability indicators constitutes a framework that requires ongoing refinement. Our hope is that this

method can contribute to generating additional knowledge that can serve as a basis for the debate surrounding Swedish and European Forestry.

Further analysis is also required to account for substitution effects of forest raw materials and products, considering different forest management techniques and the climate performances of competing products over time. We also see a large potential to include the analysis of biogenic carbon stored in wood-based products, i.e., Harvested Wood Products, further downstream the forest value chain.

**Author Contributions:** Conceptualization, P.E.K., E.M. and M.E.; methodology, P.E.K., E.M. and H.H.; validation, E.M., M.E., Å.N. and H.H.; formal analysis, P.E.K. and E.M.; investigation, E.M. and P.E.K.; writing—original draft preparation, E.M. and P.E.K.; writing—review and editing, M.E., Å.N. and H.H.; visualization, P.E.K. and E.M. All authors have read and agreed to the published version of the manuscript.

**Funding:** This research was funded by Mistra Digital Forest program under Grant DIA 2017/14.

**Informed Consent Statement:** Not applicable.

**Data Availability Statement:** The data presented in this study are available on request from the corresponding author.

**Acknowledgments:** The authors thank Emke Vrasdonk for valuable comments on the manuscript.

**Conflicts of Interest:** Authors Eskil Mattsson, Per Erik Karlsson, Martin Erlandsson and Åsa Nilsson was employed by IVL Swedish Environmental Research Institute. The remaining authors declare that the research was conducted in the absence of any commercial or financial relationships that could be construed as a potential conflict of interest.

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
