# Peer review of "Indicators of Sustainable Forestry: Methodological Approaches for Impact Assessments across Swedish Forestry"

_sustainability, doi:10.3390/su16083331_

Round 1

Reviewer 1 Report

Comments and Suggestions for Authors

The paper presents a methodological framework for integrated sustainability impact assessments using a diverse range of indicators for managed forests in Sweden.

It is an original paper, well-written, with proper structure.

The methodology is flexible, and it is explained in great detail.

The three scenarios presented in the paper add more validity to the results.

The paper's results are robust, and the conclusions are based on them.

The discussion section is well-written, providing logical explanations of the results and support them with recent literature.

The only minor weakness of the article is that it is not adequately referenced. The authors should enhance the literature review section by adding one or two more paragraphs in the Introduction.

The first paragraph could be placed after Line 58, including more relevant references for the examined topic, e.g. 

Lindner, M., Werhahn-Mees, W., Suominen, T., Vötter, D., Zudin, S., Pekkanen, M., ... & Pizzirani, S. (2012). Conducting sustainability impact assessments of forestry-wood chains: examples of ToSIA applications. European Journal of Forest Research131, 21-34.

Päivinen, R., Lindner, M., Rosén, K., & Lexer, M. J. (2012). A concept for assessing sustainability impacts of forestry-wood chains. European Journal of Forest Research131, 7-19.

Palosuo, T., Suominen, T., Werhahn-Mees, W., Garcia-Gonzalo, J., & Lindner, M. (2010). Assigning results of the Tool for Sustainability Impact Assessment (ToSIA) to products of a forest-wood-chain. Ecological Modelling221(18), 2215-2225.

The second paragraph could be placed after Line 74 including more relevant References regarding Swedish forestry.

Author Response

Point by point response to Comments and Suggestions for Authors

Comments 1: 

The paper presents a methodological framework for integrated sustainability impact assessments using a diverse range of indicators for managed forests in Sweden. It is an original paper, well-written, with proper structure. The methodology is flexible, and it is explained in great detail. The three scenarios presented in the paper add more validity to the results. The paper's results are robust, and the conclusions are based on them. The discussion section is well-written, providing logical explanations of the results and support them with recent literature.

Response 1: Thank you for this nice feedback. 

Comments 2: The only minor weakness of the article is that it is not adequately referenced. The authors should enhance the literature review section by adding one or two more paragraphs in the Introduction.

Response 2: We have now elaborated the introduction by adding additional sentences in existing paragraphs in the introduction and included more references.

Comments 3: The first paragraph could be placed after Line 58, including more relevant references for the examined topic, e.g. 

Lindner, M., Werhahn-Mees, W., Suominen, T., Vötter, D., Zudin, S., Pekkanen, M., ... & Pizzirani, S. (2012). Conducting sustainability impact assessments of forestry-wood chains: examples of ToSIA applications. European Journal of Forest Research131, 21-34.

Päivinen, R., Lindner, M., Rosén, K., & Lexer, M. J. (2012). A concept for assessing sustainability impacts of forestry-wood chains. European Journal of Forest Research131, 7-19.

Palosuo, T., Suominen, T., Werhahn-Mees, W., Garcia-Gonzalo, J., & Lindner, M. (2010). Assigning results of the Tool for Sustainability Impact Assessment (ToSIA) to products of a forest-wood-chain. Ecological Modelling221(18), 2215-2225.

Response 3: Agree. These references are valid and has been added. However, we believe the first paragraph sets the scene of the article to better extent than the second paragraph, so we kept this structure. 

Comment 4: The second paragraph could be placed after Line 74 including more relevant references regarding Swedish forestry.

Response 4: Ok - this has been changed as suggested and more references has been added.

Reviewer 2 Report

Comments and Suggestions for Authors

Indicators of sustainable forestry: methodological approaches for impact assessments across Swedish forestry

This article addresses a very current topic: integrated sustainability impact assessments using different indicators. The authors analysed in the detail methodological framework for quantifying and balancing impacts on widely different aspects of sustainability for different future scenarios for forestry.

However, several issues need to be addressed before the paper can be considered for publication:

·       the discussion is very poor in my opinion, the discussion about the achieved results, which are interesting, is missing in this chapter and it is partially included in other chapters of the article

·       Conclusions should relate directly to the results achieved and should be more detailed

After these comments are corrected, the article can be published.

Comments on the Quality of English Language

Comments on the Quality of English Language

·       The language quality is fine, but I advise before publication the article a check by a native speaker of the language.

Author Response

Point by point response to Comments and Suggestions for Authors

Comments 1: This article addresses a very current topic: integrated sustainability impact assessments using different indicators. The authors analysed in the detail methodological framework for quantifying and balancing impacts on widely different aspects of sustainability for different future scenarios for forestry.

Response 1: Thank you. 

Comments 2: However, several issues need to be addressed before the paper can be considered for publication:

The discussion is very poor in my opinion, the discussion about the achieved results, which are interesting, is missing in this chapter and it is partially included in other chapters of the article.

Response 2: Thank you for this comment. We have elaborated the discussion section where we discuss the results further including its implications and further applications in two additional paragraphs. As you correctly point out, we also discuss the results in the results section. Hence, we have merged the Results and Discussion chapters into one chapter.

Comment 3: Conclusions should relate directly to the results achieved and should be more detailed. After these comments are corrected, the article can be published.

Response 3: Good point. We have added several sentences in the conclusions to address your concern. Now it is more detailed. 

Comment 4: The language quality is fine, but I advise before publication the article a check by a native speaker of the language.

Response 4: Sentences and language has been checked and small adjustments has been made where necessary.  

Reviewer 3 Report

Comments and Suggestions for Authors

The work presented remains interesting but requires some adjustments to make it publishable.

The study should showcase the results of the evaluation/application test of the methodology in the abstract.

Introduction: In the overall text presentation, there is a lack of examples to support various assertions. For instance, concrete results obtained with TOSIA, etc. For tools like TOSIA and LECA, also highlight their strengths. In the text, it tends to emphasize their limitations more.  The introduction outlines Sweden's forest context without assessing existing evaluation methodologies, including their limitations and the need to develop a new methodology. The dominant forest type in Sweden, along with its emblematic species, isn't presented. The paper and other products made from wood should be quantified. What is the study's hypothesis?

Materials and methods: Briefly describe the site (environment) where the evaluation model is applied, specifically Västernorrland in the northern part of Sweden (boreal zone) and Kronoberg in the southern part (the boreonemoral zone). Illustrate this with a map. 

Consider presenting the three scenarios in a table and restructuring the methodology for better flow.

The interpretation of results should be concise, with a key message for readers at the end of each section.

The discussion should address methodological limitations, discuss results obtained (by comparing the two areas studied) with different indicators in detail, and provide implications for future research in Sweden.

Specific comments: The downstream parts of the forest value chains lack equivalent indicators (L99-100). Why are they not included?

Author Response

Point by point response to Comments and Suggestions for Authors

Comment 1: The study should showcase the results of the evaluation/application test of the methodology in the abstract.

Response 1: Thanks. Results has now been added in abstract.

Comment 2: Introduction: In the overall text presentation, there is a lack of examples to support various assertions. For instance, concrete results obtained with TOSIA, etc. For tools like TOSIA and LECA, also highlight their strengths. In the text, it tends to emphasize their limitations more.  The introduction outlines Sweden's forest context without assessing existing evaluation methodologies, including their limitations and the need to develop a new methodology. The dominant forest type in Sweden, along with its emblematic species, isn't presented. The paper and other products made from wood should be quantified. What is the study's hypothesis?

Response 2: Thank you for this valuable comment. We have now added the strengths of the Tosia Model. Existing evaluation models has been added including dominant forest types as well as statistics of wood, pulp and sawn-wood. Also, a hypotheses has been added.

Comment 3: 

Materials and methods: Briefly describe the site (environment) where the evaluation model is applied, specifically Västernorrland in the northern part of Sweden (boreal zone) and Kronoberg in the southern part (the boreonemoral zone). Illustrate this with a map. 

Response 3: We have now added a map (Figure 1) and added more information on environmental characteristics of the sites. We believe presenting the scenarios in a table would not give any additional information or add any more clarity. However, for better flow, the sub-chapter "Choice of indicators" has been moved to the chapter after "Forest Scenarios".

Comment 4: The interpretation of results should be concise, with a key message for readers at the end of each section.

Response 4: Good comment. Short summary results for each indicator has been added/elaborated now for each sub-chapter.

Comment 5: The discussion should address methodological limitations, discuss results obtained (by comparing the two areas studied) with different indicators in detail, and provide implications for future research in Sweden. Specific comments: The downstream parts of the forest value chains lack equivalent indicators (L99-100). Why are they not included?

Response 5: Thank you for this comment. We have elaborated the discussion section where we now discuss the results further including its implications and further applications in two additional paragraphs. Limitations concerning down-stream parts of value chain are now discussed. We have also merged the Results and Discussion chapters into one chapter.

Round 2

Reviewer 3 Report

Comments and Suggestions for Authors

I would like to thank the authors for incorporating most of my comments. However, I suggest that the results and discussion sections be written separately. 

Author Response

Thank you very much for taking the time again to review this manuscript. Please find the detailed responses below and the corresponding revisions/corrections highlighted/in track changes in the re-submitted file.

Point by point response to Comments and Suggestions for Authors

Comment 1: I would like to thank the authors for incorporating most of my comments. However, I suggest that the results and discussion sections be written separately

Response 1: Thank you once again for reading our manuscript and suggesting comments that will improve the quality of our paper. As you suggested, we have now separated the results and discussion sections.